# Bioinspired Hierarchical Carbon Structures as Potential Scaffolds for Wound Healing and Tissue Regeneration Applications

**DOI:** 10.3390/nano13111791

**Published:** 2023-06-02

**Authors:** Soham D. Parikh, Wenhu Wang, M. Tyler Nelson, Courtney E. W. Sulentic, Sharmila M. Mukhopadhyay

**Affiliations:** 1Department of Mechanical & Materials Engineering, Wright State University, 3640 Col. Glen Hwy, Dayton, OH 45435, USA; parikh.32@wright.edu; 2Frontier Institute for Research in Sensor Technologies (FIRST), University of Maine, United States Air Force Research Laboratory, Orono, ME 04469, USA; wenhu.wang@maine.edu; 3711th Human Performance Wing, Airman Systems Directorate, Bioengineering Division, Wright-Patterson Air Force Base, Dayton, OH 45433, USA; 4Department of Pharmacology and Toxicology, Wright State University, Boonshoft School of Medicine, 3640 Col. Glen Hwy, Dayton, OH 45435, USA

**Keywords:** carbon nanotubes, bioscaffolds, tissue engineering, wound healing, biomaterials

## Abstract

Engineered bio-scaffolds for wound healing provide an attractive treatment option for tissue engineering and traumatic skin injuries since they can reduce dependence on donors and promote faster repair through strategic surface engineering. Current scaffolds present limitations in handling, preparation, shelf life, and sterilization options. In this study, bio-inspired hierarchical all-carbon structures comprising carbon nanotube (CNT) carpets covalently bonded to flexible carbon fabric have been investigated as a platform for cell growth and future tissue regeneration applications. CNTs are known to provide guidance for cell growth, but loose CNTs are susceptible to intracellular uptake and are suspected to cause in vitro and in vivo cytotoxicity. This risk is suppressed in these materials due to the covalent attachment of CNTs on a larger fabric, and the synergistic benefits of nanoscale and micro-macro scale architectures, as seen in natural biological materials, can be obtained. The structural durability, biocompatibility, tunable surface architecture, and ultra-high specific surface area of these materials make them attractive candidates for wound healing. In this study, investigations of cytotoxicity, skin cell proliferation, and cell migration were performed, and results indicate promise in both biocompatibility and directed cell growth. Moreover, these scaffolds provided cytoprotection against environmental stressors such as Ultraviolet B (UVB) rays. It was seen that cell growth could also be tailored through the control of CNT carpet height and surface wettability. These results support future promise in the design of hierarchical carbon scaffolds for strategic wound healing and tissue regeneration applications.

## 1. Introduction

Since 2008, approximately five million cases of traumatic injuries involving skin (accidents, burns, firearms, intensive surgeries) have been registered every year, with a financial burden of $25 billion annually in the USA alone [1], not counting the additional millions of cases and related expenses worldwide [2]. Current therapies for wounds and burns include wound debridement, infection management, surgical dressings [1], and skin grafts/substitutes, all of which address immediate symptoms [3]. Surgical intervention represents the current gold standard of care and is an appropriate treatment modality. However, this option can pose challenges in terms of limited availability of donor skin grafts (auto-, allo-, or xeno-grafts), donor (allo- or xeno-graft options) related immune rejection and integration of grafts, limited mechanical integrity, infection management, and burden of post-operative care placed on the patient or caregiver [4,5,6,7]. Although in the early stages of development, engineered skin may become a promising alternative treatment in the future. Modified fiber mats can be used to derive autologous skin through the integration of primary cells and scaffolds. Such engineered skin would be grown in the laboratory by seeding a biocompatible material with skin-representing cells and matured before being applied as a graft or enhanced surgical dressing [8]. In the context of the current study, we define biocompatibility as supporting cellular proliferation without significantly inducing cytotoxicity or an overt immune response. Despite showing potential advantages over conventional wound management treatment options, such as reduced dependence on donors, faster repair rate, and better tissue remodeling, bioscaffolds are far from being generalized treatment options for wound healing due to their limitations in handling, preparation, sterilization, shelf life, and effectiveness in varied wound types [9]. There is a need for a more robust, sterilizable, and easy-to-handle bioscaffold with a longer shelf life for rapid wound healing [10]. In general, there are four major classes of bio-scaffolds: natural biological materials, synthetic materials, composites, and nanostructured solids, each of which provides unique advantages depending on the targeted application [11,12,13]. A growing trend is to use nanostructured bio-scaffolds as they allow detailed surface manipulation and structural hierarchy that can mimic the microstructure and morphology of the native tissue [13,14], providing a more direct pathway for bioengineered materials to replace the functionality of the original tissue [15,16].

Carbon nanotube (CNT)-based scaffolds are becoming popular [17] because of their unique potential advantages. CNTs are relatively easy to manufacture, functionalize [18], and sterilize [19]. They also have robust mechanical properties [20], elasticity [21], and electrical conductivity. They are potentially biocompatible [22,23,24], can be functionalized with different growth factors and molecules [25,26], and are easy to sterilize [27]. CNT functionalization can be guided to reduce graft rejection by altering their cellular interaction pathways, rendering the material biocompatible [22,23,24]. CNTs also have biophysical contact cues such as electrical conductivity and surface nanoroughness, which can be utilized to control and regulate cell growth, migration, and aid in cell directionality [28,29,30]. The surface wettability of CNT scaffolds is tailorable [31,32] and can be adjusted to facilitate cell adhesion and cell growth [33]. Due to such properties, the utility of CNTs has been investigated in a variety of biomedical applications such as drug delivery systems, gene therapy, biosensors, theragnostic applications, orthopedics, and many more [34,35,36].

Despite the unique advantages of CNTs for regenerative medicine, the major limitation of existing CNT-based studies is that they involve detached or loosely aggregated CNTs that can also cause in vitro and in vivo cytotoxicity [37,38,39,40]. While the toxicity of individual CNTs may depend on purity, functionalization, and dimension, the overall toxicity associated with loose CNTs in the biological environment can still be unpredictable and often lead to adverse effects [41,42,43]. To address this concern, the current study evaluated the biocompatibility of a robust and durable bioscaffold structure where CNT arrays are covalently conjugated to flexible and biocompatible carbon fiber cloth. Earlier studies on these materials for other applications have shown that these bonds are strong enough for the nanotubes to remain attached to the surface even in rapidly flowing and agitated fluid environments [44,45,46]. For example, a CNT scaffold was subjected to extreme agitation with the intent to fracture the material [45]. The top layer of the graphitic substrate peeled off from the adjacent graphitic layers, but the CNT nanocarpets stayed attached to the top layer of the graphitic substrate. These results indicate that the CNT-substrate bonding is stronger than the inter-graphitic bonds of the underlying solid. This approach would capitalize on the functionalizable topological features and extensive surface area of CNTs for tissue engineering applications while mitigating their potential cytotoxic effects.

In the present study, CNT-coated scaffolds with tailored length as well as wettability were analyzed to assess keratinocyte and fibroblast cell growth potential as a proof of concept for future skin or tissue grafting applications such as large wounds. Aligned, 2D flexible carbon fiber mats were modified by growing carbon nanotubes through a two-step process of surface plasma activation followed by chemical vapor deposition (CVD). Due to the covalent nature of CNT bonding, toxicity risks associated with free CNTs can be avoided, as described above [44,45,46]. The carbon fiber mats are also available in various morphologies and weave patterns. Therefore, the starting substrate can be selectively chosen according to tissue characteristics; CNT length and density can also be controlled. By choosing the right combination of substrate and CNT carpet parameters, a hierarchical CNT-coated scaffold can be tailored on both nano and micro scales. Our studies indicate that CNT-coated scaffolds are supportive of keratinocyte and fibroblast migration and that the length and surface functionality of CNTs has a significant effect on keratinocyte cell growth.

## 2. Materials and Methods

### 2.1. Preparation and Surface Treatment of CNT-Coated Scaffolds

Carbon fiber cloth was enhanced with covalently bound carbon nanotubes (CNT) using a two-step chemical vapor deposition (CVD) method as described in an earlier publication [47]. The first step is to use a microwave plasma-enhanced chemical vapor deposition (PECVD) in a plasma reactor (V15GK, PlasmaTech Inc., Carson City, NV, USA) to deposit a silica buffer layer on a carbon fiber cloth. This buffer layer-coated substrate is followed by thermally activated chemical vapor deposition (CVD) of aligned carbon nanotubes using a floating catalyst technique via a three-zone programmable furnace system (OTF 1200X-III, MTI Corporation Ltd., Richmond, CA, USA). Detailed studies of CNT growth using this approach, as well as interatomic bonds in this structure, are reported in earlier publications [44,47,48]. For this study, the deposition times of the thermal chemical vapor deposition step were varied, as described in the results.

Two types of scaffolds, carbon fiber supports with covalently attached CNTs (CNT-coated) and bare carbon fiber cloth (CFC), were compared. Prior to cell seeding, all scaffolds were pre-treated with methanol and 1 M nitric acid, followed by treatment with 70% ethanol. When the scaffolds were completely dried, they were autoclaved for 30 min at 121 °C, followed by incubation in 70% isopropanol. 

To understand the effect of surface wettability, CNT-coated scaffolds that are naturally hydrophobic were modified to be hydrophilic. This was achieved by incubating the scaffolds for 1 h with 10% sodium hypochlorite-based bleach, followed by repeated washing with autoclaved DI water. Once dried, all the hydrophilic scaffolds were pre-wetted overnight with complete media to aid cell attachment [49]. 

### 2.2. Structure Analysis of Scaffolds Using Scanning Electron Microscopy (SEM)

SEM was performed in a JEOL 7401F field emission scanning electron microscope (FE-SEM). Scaffolds were air-dried and sputter coated with gold particles (Denton Desk V, 10 s at 50 mA). The scaffolds were visualized on SEM using ~3 kV accelerating voltage and a working distance of ~8 mm. Energy dispersive X-ray spectroscopy (EDS) data were collected using a probe current of 12 mA and an accelerating voltage of 20 kV. Data from at least two different locations per sample were averaged and quantified.

### 2.3. Analyses of Surface Chemical States Using X-ray Photoelectron Spectroscopy (XPS)

XPS on CNT-coated samples was carried out using the Kratos (Axis Ultra; Heywood, UK) system and was performed in an ultra-high vacuum environment (~10^−9^ Torr) with a monochromatized Al Kα (1486.6 eV) source. These acquired data were processed with Casa XPS software (Casa Software Ltd., Heywood, UK). Data analysis was carried out by using duplicate samples, selecting multiple spots in each sample, and optimizing the scan location for maximum output counts. 

### 2.4. Cell Culture and Cell Seeding on Scaffolds

Immortalized keratinocytes (HaCaT) were originally obtained from Petra Boukamp, German Cancer Research Center, Heidelberg, Germany [50]. Telomerase immortalized, normal human foreskin fibroblasts (NHF1) were originally obtained from Dr. William Kaufmann at the University of North Carolina. Both cell lines were maintained in a complete medium of HyClone Dulbecco’s Modified Eagle Medium supplemented with high glucose (DMEM; GE, Chicago, IL, USA), supplemented with 10% Fetal Bovine Serum (FBS; Corning, VA, USA), 1% L-Glutamine (200 mM; Gibco, NY, USA), and 1% Antibiotic-Antimycotic (Gibco, NY, USA). Both cell lines were maintained at ~80% confluence. HaCaTs were split with 0.025% Trypsin–EDTA (0.25%; Gibco, NY, USA), and NHF1s were split using 0.05% Trypsin–EDTA (Gibco, NY, USA). Fresh culture media was provided every 48 h. 

Suspension cell lines CL-01 and SKW 6.4 were used for B cell studies. These cells were derived from different Burkitt lymphoma patients and are EBV-transformed cell lines. The CL-01 cell line (Novus Biologicals, CO, USA) expresses surface IgM and IgD and can undergo class switch recombination upon stimulation as previously characterized [51,52]. SKW 6.4 cells (SKW, ATCC^®^ TIB 215™, VA, USA) were derived originally from the Daudi B-cell line [53] and express surface IgM. Both cell lines were maintained in a complete medium of RPMI 1640 media (HyClone, GE, Chicago, IL, USA) supplemented with 10% bovine calf serum (Thermo Fisher Laboratories, Waltham, MA, USA), 13.5 mM HEPES (Sigma, St. Louis, MO, USA), 1.0 mM non-essential amino acids (Corning, VA, USA), 1.0 mM sodium pyruvate (Corning, VA, USA) and 50 μM 2-mercaptoethanol (Fisher, UK). The pH was adjusted using filter-sterilized 1 N NaOH. CL-01 cells were maintained around 1 × 10^5^ cells/mL to 3 × 10^5^ cells/mL, and SKW cells were maintained around 2 × 10^5^ cells/mL to 1 × 10^6^ cells/mL. Human interleukin-4 (IL-4, Cell Signaling Technology Inc., Danvers, MA, USA) and human recombinant CD40 ligand (MEGACD40L^®^, Enzo Life Sciences, Farmingdale, NY, USA) were used to stimulate the B cells. All cell lines were incubated and maintained at 37 °C in a humidified 5% CO_2_ environment. 

### 2.5. Cell Seeding with CNT-Coated Scaffolds

For the adherent cell lines (HaCaT and NHF1), 100–150 μL of a concentrated cell suspension was seeded on top of the scaffold surface. Seeding densities were 5 × 10^4^ cells/well for 12-well plates, 2.5 × 10^4^ cells/well for 24-well plates, 0.5 × 10^4^ cells/well for 96-well plates, and 1.5 × 10^5^ cells/plate for 35 mm^2^ plates unless otherwise mentioned. The scaffolds were then incubated for 4 h to allow cellular attachment. After 4 h, culture media was added to the plates. To understand how non-adherent cells interact with CNT-coated scaffolds, SKW and CL-01 cells were first seeded in 12-well plates at 2 × 10^4^ cells and 1 × 10^5^ cells in a 2 mL suspension, respectively. After 24 h, scaffolds were introduced to each well. 

### 2.6. Elution Test

The simulated wound fluid (SWF) used in the study was made according to the formula by Bradford et al. [54]. Briefly, 0.1 M sodium chloride, 40 mM sodium hydrogen carbonate, 4 mM potassium chloride, 2.5 mM calcium chloride, and 3% bovine albumin were added to 100 mL deionized water to prepare the SWF. To prepare the extracts for the elution test, scaffolds were incubated in SWF for 14 days. These prepared extracts were first checked for their pH and further diluted to 25%, 50%, and 75% in either cell culture media or quiescent cell culture media (i.e., 1% FBS). 

For the elution test, the HaCaTs were grown in 96-well plates until they were confluent. Quiescent cells were prepared by first incubating the cells in culture media until the cells reached near confluency and then culturing the cells for 24 h with quiescent media (1% FBS) [55,56,57]. Normally growing and quiescent cells were incubated for 1 or 3 days in the presence of extracts prepared from the scaffolds. Cell growth was assessed visually by microscopy, and the cytotoxicity was measured with the CytoTox ONE™ Homogeneous Membrane Integrity Assay (Promega, WI, USA) according to manufacturer instructions on days 1 and 3. 

### 2.7. Cell Labeling 

When noted, cells were stained with cell tracking dye, CellTrace™ CFSE (Thermo Fisher, MA, USA), using the manufacturer’s instructions. Briefly, for labeling, the cells were trypsinized, counted, and the cell suspension was centrifuged. The pellet was resuspended in 1× PBS with dye staining solution, and cells were incubated in the dark for either 15 min for CFSE or 1.5 h for Calcein AM (Thermo Fisher, MA, USA) and mixed with an equal volume of complete culture media. After the incubation, the suspension was again centrifuged, pellets were resuspended in the complete media, and plating was performed. For fixing Calcein AM stained cells with paraformaldehyde, cells were first washed 2 times with PBS and incubated with 4% paraformaldehyde for 1 h. For nuclear staining, cells were first washed 2 times with PBS and stained with 4′,6-diamidino-2-phenylindole (DAPI) for 10 min. 

### 2.8. Ultraviolet-B (UVB) Treatment

CellTrace™ CFSE stained cells were seeded on carbon bioscaffolds in 12-well plates at 0.25 × 10^6^ cells/scaffold. After allowing 24 h for cell attachment, Long CNT-coated hydrophilic scaffolds, Short CNT-coated hydrophilic scaffolds, or CFC were first imaged using an Agilent BioTek Cytation5 cell imaging multi-mode reader by taking 6×6 montage images at 4× magnification. These images were used as individual controls for each condition for cell counts. The scaffolds were then transferred to a 35 mm plate, covered with 0.5 mL 1× PBS, and treated with 100 J/m^2^ UVB using a Philips TL 20 W/12 RS SLV/25 light. Before UVB treatment, culture media was replaced with 400 μL of 1× PBS as UVB treatment with culture media can induce cytotoxicity [58]. The lids of the culture plates were also removed prior to UVB exposure to ensure direct exposure of the cells. Immediately after treatment, the cells were washed two times with PBS, and then 2 mL of complete media was added to each 35 mm^2^ plate. Cell viability was calculated using the before and after pictures of individual scaffolds per group.

### 2.9. Cell Proliferation 

Cell proliferation of cells seeded on the various carbon scaffolds was determined using the CellTiter 96^®^ Aqueous One Solution Cell Proliferation Assay kit (Promega, WI, USA). After the cells were grown for either one, three, or five days, samples were washed twice with 1× PBS and incubated in MTS colorimetric reagent for 1.5 h. After the incubation, 100μL of the solution was transferred to a 96-well plate, and absorbance was measured at 490 nm using an Agilent BioTek Synergy H1 hybrid multi-mode fluorescence microplate reader. Cell proliferation was expressed as the normalized values against the control wells containing scaffolds with the culture medium without cells. Cell culture media was replaced every two days. 

### 2.10. Cytotoxicity Analysis 

Cytotoxicity was evaluated using the CytoTox ONE™ Homogeneous Membrane Integrity Assay reagent (Promega, WI, USA), which measures LDH release. HaCaTs were grown on different scaffolds for one, three, or five days and culture media was exchanged every two days. At each incubation time point, plates were first incubated at room temperature for 20 min, and the supernatant was collected and mixed with an equal volume of LDH assay reagent mix according to manufacturer instructions. After 15 min of incubation, the reaction was stopped, and fluorescence was measured at 560 nm/590 nm using an Agilent BioTek Synergy H1 hybrid multi-mode fluorescence microplate reader. The LDH positive control was used to lyse cells and obtain the maximum possible LDH release. 

### 2.11. Cell Migration from CNT-Coated Scaffolds

To understand cell migration from the scaffolds, all the carbon bioscaffolds used in the study were placed in a 12-well plate and seeded with 1 × 10^6^ cells/scaffold for HaCaTs and 0.5 × 10^6^ cells/scaffold for NHF1 cells. After 24 h incubation, the scaffolds were washed twice with 1× PBS and transferred to an empty well, and fresh media was added to the scaffolds and incubated for up to 10 days. After the incubation period, cell migration onto the empty plate was assessed via brightfield imaging. To analyze cell migration towards the scaffolds, 0.75 × 10^5^ NHF1 or 1.5 × 10^6^ HaCaT cells were seeded per well in 12-well plates, and 24 h after incubation, scaffolds were placed in the wells with pre-seeded cells. The scaffolds were removed on day 5 and stained with Calcein AM, fixed with 4% paraformaldehyde, stained with DAPI, and visualized using an AMEFC4300 EVOS™ microscope.

### 2.12. Sandwich Enzyme-Linked Immunosorbent Assay (ELISA)

ELISA was performed as described previously [59] to measure IgM and IgG secretion from B cell lines. For analyzing antibody secretion, the cell suspension was collected, centrifuged at 680× *g* for 5 min, and the supernatant was collected and stored at −80 °C for later ELISA analysis. Briefly, 96-well ELISA plates were coated with 100 μL/well of either goat anti-human IgG Fc (ab97221, Abcam, Waltham, MA, USA) at 1:500 or goat anti-human IgM (2010-01, SouthernBiotech, Birmingham, AL, USA) at 1:1500 and incubated overnight at 4 °C. Dilutions were made using 0.1 M sodium carbonate bicarbonate buffer, pH 9.6. Following the overnight incubation, plates were washed twice with 1× PBS + 0.05% Tween-20 and thrice with ddH_2_O and then blocked overnight at 4 °C or for 2 h at room temperature with 200 μL/well of blocking buffer consisting of 1× PBS + 3% Bovine Serum Albumin (BSA, Millipore Sigma, St. Louis, MO, USA). Plates were again washed twice with 1× PBS + 0.05% Tween-20 and thrice with ddH_2_O. 

Standards for IgG (A80-105, Bethyl Laboratories, Montgomery, TX, USA) and IgM (0158L-01, SouthernBiotech) were diluted in RPMI complete media, and 100 μL of standards were added to the plates along with 100 μL of supernatant from each sample. Plates were incubated for 1.5 h at 37 °C and then washed thrice with PBS + 0.05% Tween-20 and four times with ddH_2_O. HRP conjugated goat anti-human IgG (A80-104P, Bethyl Laboratories) diluted at 1:10,000 or HRP conjugated goat anti-human IgM (2020-05, SouthernBiotech) diluted at 1:4000 was added to each well (100 μL/well) and incubated for 1.5 h at 37 °C. Plates were then washed thrice with PBS + 0.05% Tween-20 and four times with ddH_2_O. Finally, 100 μL TMB substrate (Millipore Sigma) was added to each well and incubated for 30 min in the dark for IgG or 5 min for IgM. The stop solution (4N H_2_SO_4_, Fisher Scientific, Morris Plains, NJ, USA) was added (100 μL/well) after the appropriate time frame, and the absorbance values were measured at 450 nm using a SpectraMax Plus 384 UV/VIS microplate spectrophotometer (Molecular Devices, San Jose, CA, USA). SoftMax Pro was used to process the standard curves and calculate the secreted IgG and IgM.

### 2.13. Cytokine Analysis

Using a 25-Plex Human ProcartaPlex™ Panel 1B (Invitrogen, Austria), cytokines were measured in cell culture supernatants of HaCaTs grown for 5 days on the scaffolds as well as SKW and CL-01 cells grown in the presence of the scaffolds for 4 days. Reads were performed using a Luminex FLEXMAP 3D^®^ System.

### 2.14. Data Analysis

All experiments were performed 2–3 times with sample triplicates. Results from triplicates were averaged and counted as a single n for statistical analysis. Data are reported as the mean ± standard error of the mean (SE). Statistical analysis was carried out using GraphPad Prism. Statistical significance between the CNT and pristine groups was determined by one-way ANOVA and Tukey post hoc analysis. Statistical significance in the fold-effect between scaffold groups and the controls was determined by two-way ANOVA analysis followed by Bonferroni post-tests or Dunnett’s multiple comparisons test and comparing each sample to the corresponding cell controls and/or CFC controls. Statistical significance in the fold effect between UVB and control groups was determined by one-way ANOVA with Tukey post hoc analysis comparing all groups to cell controls. Statistical analyses were performed with a 95% confidence interval (*p* < 0.05) for all the ANOVA tests performed above.

## 3. Results

### 3.1. Surface Morphology of the CNT-Coated Carbon Fiber Cloth 

Carbon fiber cloth before and after enrichment with covalently bonded CNT arrays was visualized by a scanning electron microscope (SEM) (Figure 1). Details about processing techniques and parameters to control structure and surface chemistry have been elaborated in earlier publications [47,60]. For the CNT-coated scaffolds, the strands of nanotubes are vertically aligned near the substrate but start entangling at the top as the nanotubes grow longer (Figure 1B). From in-depth surface morphology experiments reported earlier [61], the mean inner diameter of individual nanotubes is about 9.9 (±0.76) nm, and the mean outer diameter is about 22.4 (±1.84) nm. It was also noted that after the first few minutes of initial nucleation and growth, the CNT layer covers most substrates uniformly. Because of this, CVD run time determines the overall length of the CNT carpet.

To understand the influence of CNT carpet length on cell proliferation, two deposition times were carefully selected in the CVD process. A short time point of 20 min was selected to achieve uniform nucleation, even CNT growth, and a shorter CNT carpet length and 120 min was chosen for obtaining long CNT carpets. Based on previous analysis, the CNT lengths are approximately 13–16 μm and 190–230 μm for the 20 min and 120 min samples, respectively [61]. Detailed surface area analyses were previously performed using gas adsorption isotherms, and the specific surface areas were approximately 1.5 m^2^/g for the 20 min sample and about 11.5 m^2^/gm for the 120 min sample [60].

Nanotubes were observed to be uniformly distributed throughout the fiber cloth substrate, and a characteristic CNT carpet covering the entire surface of the substrate was observed (Figure 1C–H). Consistent with estimates made from earlier studies [60], uniform but significantly less dense CNT carpets are seen following a 20 min coating, referred to as ‘Short CNTs’ due to their shorter lengths (Figure 1C,E,G). The 120 min coating produced longer, more densely entangled CNT carpets, referred to as ‘Long CNTs’ (Figure 1D,F,H).

### 3.2. Influence of Surface Oxidation Treatment on Surface Chemical States of CNT Carpet

X-Ray Photoelectron Spectroscopy (XPS) was obtained from the CNT-coated material before and after surface oxidation treatment. XPS peaks represent the binding energy of atoms in the outermost atomic layers (1–10 nm) of the solid, hence indicative of how the chemical states of surface atoms may be impacted by any given treatment.

An XPS survey scan provides a low-resolution outlook of all the surface elements and was performed on the samples before and after hypochlorite bleach (surface oxidation) treatment for the pure CNT-coated sample; only the carbon (C1s) peak is visible with binding energy 285 eV, which is characteristic of pure graphitic carbon [62] (Figure 2). This agrees with earlier reported studies [63,64,65] on CNT-coated carbon. Oxidation treatment of the CNT–CFC sample using bleach treatment adds a peak correlating with oxygen (O1s peak) at 533 eV in addition to the carbon (C1s) peak at 285 eV (Figure 2B). This can be attributed to the attachment of oxygen-containing functional groups caused by the surface oxidation treatment. This type of functionalization obtained by other treatments in the past [65] has resulted in making CNT surfaces hydrophilic. In this study, the functionalization obtained through bleach treatment, common in biomaterial usage, also has the same effect, as shown in the next section.

No traces of chlorine (Binding energy 199–201 eV) were seen on this spectrum, indicating that the washing steps after surface treatment resulted in effective rinsing and successful elimination of any remnant surface moieties from the bleaching agent. 

### 3.3. Influence of Surface Oxidation Treatment on Surface Wettability and Water Contact Angle

Earlier studies [32,66] have shown that the CNT-coated scaffolds are inherently superhydrophobic, exhibiting water contact angles of 150° and above. In this study, we investigated the influence of surface oxidation of these samples with bleach treatment to make them more hydrophilic. Water droplets were imaged on the scaffold surface before and after sodium hypochlorite-based bleach treatment (Figure 3). It was seen that the untreated, CNT-coated scaffolds are superhydrophobic with contact angles of above 150°, which was consistent with our previous studies [32]. With sodium hypochlorite-based bleach treatment, the modified hydrophilic surface soaks up any water droplet within 10 s; hence the contact angle becomes 0°, as shown in Figure 3, similar to our previous findings [66].

### 3.4. CNT-Coated Scaffolds Do Not Induce Cytotoxicity in Keratinocytes 

Carbon nanotubes, often used as loose and freely mobile entities, have been reported to induce cellular toxicity [37,38,39,67]. To determine the biocompatibility and safety of our scaffolds containing strongly anchored CNT arrays, we followed the FDA and U.S. Pharmacopoeia guidelines [68] by first performing an elution test, which measures the indirect cytotoxicity. The scaffolds were incubated in simulated wound fluid (SWF) for 14 days. SWF was used to imitate the natural wound-healing process [54,69]. After incubation, the SWF extracts were isolated from the scaffolds, and 25%, 50%, and 75% of the SWF extracts were incubated for 1 or 3 days with HaCaT cells grown normally or under quiescent conditions. Quiescent cells are less prone to toxic effects and stress [70,71]. Therefore, we evaluated if the scaffolds are able to potentiate toxic effects that could supersede the stress-resistant nature of quiescent cells.

There was no change in the pH with any of the SWF extracts. No noticeable differences in HaCaT morphology were visualized by brightfield microscopy for any extract concentrations on Day 1 or 3. There was also no significant increase in cytotoxicity as measured by LDH release at any extract concentrations on either Day 1 or 3 (Figure 4).

By adhering to non-standard direct contact study guidelines as per ISO 10993-5 by FDA [72], we evaluated potential cell cytotoxicity caused due to direct contact with scaffolds. As measured by LDH release, no direct cytotoxicity of HaCaT cells was detected on Days 1, 3, or 5 for any of the bioscaffolds (Figure 5).

### 3.5. Short CNTs Can Effectively Support Keratinocyte Cell Proliferation

To determine the effects of CNT length as well as surface wettability of the nanomaterials on cell growth of keratinocytes, we measured cell proliferation of HaCaTs on the hierarchical scaffolds compared to traditional cell culture (cell control group) via MTS assay. From short-term growth on Day 1 (Figure 6), we observed that the cell proliferation on various scaffolds as well as the cell control group was not significantly different. This was consistent with our previous observations [66], and the relatively slower proliferation rates could be associated with the substantially high surface area and the 3D nanostructure provided by the CNTs, which can slow down cell attachment compared to flat, tissue culture plates. From the 3-day cell proliferation results, we observed that cell proliferation between the CFC group and Short CNT-coated scaffolds was comparable and not significantly different (Figure 6). On Day 5, we found that cell proliferation between the cell control group, CFC group, and Short CNT-coated hydrophilic scaffolds was comparable and not significantly different. This implies that cell growth was well supported on Short CNT-coated hydrophilic scaffolds compared to their hydrophobic counterparts. Both hydrophilic and hydrophobic Short CNT surfaces, similar to CFC and cell controls, also showed higher cell proliferation as compared to the Long CNT-coated scaffolds (Figure 6). Therefore, for keratinocytes, shorter CNT coatings may be better suited as bioscaffolds, and the hydrophilic scaffolds are more effective at promoting cell proliferation than hydrophobic scaffolds. 

### 3.6. CNT-Coated Scaffolds Support Cell Migration

Keratinocyte and fibroblast migration and proliferation are key steps involved in the wound-healing process [73]. Hence, the effect of the scaffolds on cell migration was assessed. Since hydrophilic scaffolds were more effective for cell growth, only those were tested. HaCaT (keratinocyte) and NHF1 (fibroblast) cells migrated from a Short CNT-coated scaffold (Figure 7A,B). Similar to the Short CNT-coated scaffolds, HaCaT and NHF1 cells migrated from the CFC and Long CNT scaffolds. To further validate the cell migration findings, we performed a study to determine if the cells grown on the culture plates can migrate onto the scaffolds. Both keratinocytes and fibroblasts migrated onto the Short CNT-coated scaffolds (Figure 7C,D). Similar migration was observed on Long CNT-coated scaffolds and CFC. Cells can effectively migrate towards the CNT-coated scaffolds and through the leading weaves and fibrils. We also observed the migrated cells towards the central region of the scaffolds and throughout the scaffold surface, implying that once attached, the cells can continue migrating on these CNT carpets. These results suggest that these scaffolds could support wound healing at the site of injury. Additionally, the scaffolds appear to provide directionality of the migrating keratinocytes and fibroblasts, which is a property that can be exploited in wound healing scaffolds.

### 3.7. Scaffolds Provide Cytoprotection against UVB Exposure

The flexibility and favorable cell growth properties of CNT-coated scaffolds support their potential application as a split-thickness skin graft in large wounds. We hypothesized that the scaffolds would be cytoprotective against sun exposure and burn. Therefore, HaCaT cells grown on CNT-coated scaffolds were exposed to UVB treatment. We first tested HaCaT cells against various doses of UVB, and 100 J/m^2^ was the lowest dose that induced significant cell death in HaCaTs and was thus chosen for the study. Since hydrophilic scaffolds were more effective for cell growth, only those were tested. All the scaffolds, including the CFC control, were protective against both indirect and direct UVB treatment (Figure 8). Additionally, the Long CNT-coated scaffolds exhibited significantly increased cells compared to the cell control (Figure 8). Such scaffolds, due to their longer CNTs and increased growth times of CNTs, provide dense nanotube carpets and provide extensively intertwined nanostructures for cell growth. Such scaffolds, due to their structural conformation and increased CNT area, could provide a greater blockade against UVB compared with Short CNTs, and this may explain their potential increased cytoprotective effects.

### 3.8. CNT-Coated Scaffolds May Modulate Immune Function

To gain insights into the potential immunogenic nature of the CNT-coated scaffolds, we evaluated the effect of the scaffolds on B-lymphocyte function using two human B-cell lines (SKW and CL-01). Upon cell counting after a 4-day incubation with the scaffolds, we did not observe any significant changes in cell number compared to the control group. However, we observed an increasing number of cells interacting and attaching to the CNT-coated scaffolds through phase contrast microscopy. Such bio-nano interactions could potentially alter the cellular function of these cells. Therefore, antibody secretion from both B-cell lines was evaluated after incubation with the scaffolds for 4 days. B lymphocytes must be activated to induce antibody secretion; thus, both cell lines were stimulated with CD40L and IL-4, which have been previously shown to induce B-lymphocyte activation and antibody secretion [51,74].

Since hydrophilic scaffolds were more effective for cell growth, only those were tested. In initial laboratory studies (unpublished work), CL-01 cells were observed to show a two-fold increase in IgG secretion and a slight increase in IgM secretion. With these scaffolds, we observed a similar effect: (Figure 9A). Interestingly, all scaffolds showed enhanced IgG secretion, while only the Long CNT-coated scaffolds showed significantly enhanced IgM secretion (Figure 9B).

The effect of CNT-coated scaffolds on antibody secretion was evaluated in another human B-cell line (SKW) derived from a different patient. Unlike the CL-01 cells, CD40L and IL4 stimulation induces IgM but not IgG secretion from SKW cells, perhaps due to the two B-cell lines being at different maturation stages. In contrast to the CL-01 cell line, the scaffolds had no impact on IgM secretion in the SKW cell line (Figure 9C). These results suggest that antibody secretion may be modulated depending on the B-cell maturation stage and should be evaluated when designing a bioscaffold-based implant.

To further investigate the immune function of the scaffolds, the effect of hydrophobic and hydrophilic scaffolds on cytokine profiles was evaluated in the HaCaT cells and the B-cell lines. For this assay, a Luminex 25-Plex Human cytokine panel (ProcartaPlex™ Panel 1B, Invitrogen) was used. CFC scaffolds significantly induced secretion of IL-6, IL-12, GM-CSF, and TNF-α from HaCaT cells after a 5-day incubation (Figure 10A). Long CNT-coated hydrophobic scaffolds significantly increased IL-4 secretion from HaCaT cells. Both hydrophilic and hydrophobic Short CNT-coated scaffolds significantly increased GM-CSF secretion. Short CNT-coated hydrophobic scaffolds also increased IL-17a, and Short CNT-coated hydrophilic scaffolds significantly increased IL-12 levels. In general, Short CNT-coated scaffolds had a modest effect on the secretion of inflammatory cytokines from HaCaT cells, and Long CNT-coated scaffolds had no effect (Figure 10A).

The cytokine secretion profile from the CL-01 and SKW B-lymphocyte cell lines was evaluated with or without stimulation after a 4-day incubation with the scaffolds (Figure 10B). For the CL-01 cells, only IL-4 showed any variation in secretion. IL-4 secretion was significantly increased in the presence of CFC control and trended toward an increase with Long and Short CNT-coated hydrophilic scaffolds in naïve, unstimulated CL-01 cells. However, in stimulated CL-01 cells, IL-4 was significantly reduced in cells exposed to Short CNT-coated hydrophilic scaffolds. With the SKW cells, IL-4 and IL-10 were the only cytokines modulated by exposure to the scaffolds. IL-10 secretion significantly increased in SKW cells exposed to Short CNT-coated hydrophilic scaffolds but was significantly decreased with the CFC control as compared to cell controls. For stimulated SKW cells, IL-4 secretion was significantly reduced in cells exposed to Short CNT-coated hydrophilic scaffolds, which was similar to the CL-01 cells. Additionally, IL-10 secretion significantly increased in cells treated with carbon fiber cloths. Overall, the changes in cytokines are relatively minimal and do not suggest an autocrine effect since these cytokine changes do not correspond to the effects of the scaffolds on antibody secretion.

## 4. Discussion

As mentioned earlier, carbon nanotubes (CNT) promise unique advantages for regenerative medicine, but one of the major limitations is that free-standing CNTs are susceptible to intracellular uptake and present risks of in vitro and in vivo cytotoxicity [37,38,39,67]. The current study addresses this issue by creating covalently bonded carpet-like arrays of CNT on larger substrates that can be selected based on the structural requirements of the targeted application area. This prevents the risk of unintended intake of loose nanoscale materials and may reduce any subsequent induction of autophagy. Such hierarchical tissue scaffolds could provide the advantages of nanoscale cellular and molecular interactions without the potential toxicity risks associated with isolated, non-bound nanomaterials. A tissue engineering scaffold should possess a combination of chemical compatibility and biophysical characteristics that enable the proliferation, reproduction, and regeneration of targeted cells (keratinocytes and fibroblasts in this study). For chemical compatibility, carbon may be the most suitable material since it is the basic constituent of living tissues and therefore expected to be well-tolerated. Biophysical characteristics of importance for grafts include a fibrous microstructure of suitable mechanical strength, nano-roughness for improving cell-scaffold interaction and cell adhesion, and an electrically conductive surface for cell signaling. In our materials, the carbon microfibers provide the base structural support of the scaffold. The CNTs covalently attached to these fibers provide the necessary nano-roughness as well as substantially increased surface area for cell growth, a highly tailorable surface for functionalization, and electrical conductivity. This creates a hierarchical structure that synergizes the advantages of different scales, as often seen in native biological tissues. The carbon fiber mats are also available in various morphologies and weave patterns. Therefore, the starting substrate can be selectively chosen according to the tissue characteristics. 

It is seen that cell growth on the scaffold can be modulated by a wide variety of factors, such as material surface roughness, composition, conformation, orientation, functionalization, wettability, surface area, and porosity. Surface roughness in compact solids indicates the relative interface area between solid and liquid compared to an ideal flat surface and is one of the factors that can regulate cell adhesion. For our hierarchical scaffolds, the roughness values can be modulated to a very great extent by controlling the CNT morphology and distribution, and we have previously demonstrated that nanoroughness achieved through CNT surfaces improves cellular attachment [33,75]. Scaffold wettability or hydrophilicity is another factor that can influence cell adhesion and cell growth. Previous work with glioblastoma (GBM) cells showed a slowed initial growth pattern on superhydrophobic covalently-linked CNT scaffolds [66]. However, once the covalently linked CNT surfaces were functionalized to be hydrophilic, this alteration in wettability resulted in a standard growth curve pattern instead of the initial slow growth patterns noted in non-treated superhydrophobic scaffolds [66]. This effect appeared due to increased cell adhesion capability [66]. In contrast, the effect of wettability appeared to be a minor player in the proliferation of a keratinocyte cell line (HaCaT). The differences in response to wettability by two different cell types (i.e., GBM and HaCaT) imply that cellular proliferation on these nanocarpets is cell type-specific and needs to be thoroughly evaluated before using them as implant materials.

The length of the nanotubes in the carpet was seen to have a significant effect on cell growth. Whereas numerous studies have assessed the toxicity profiles of CNTs based on their lengths [76,77,78], very little is known about if and how the length of CNTs may modulate cell proliferation. Depending on their size and functionalization, CNTs show varying degrees of *in vivo* distribution and cytotoxicity [78]. Some studies have suggested that free, Short CNTs are more effective for cellular growth and less cytotoxic due to their potential internalization and removal compared to longer CNTs or clusters of CNTs, which cannot be engulfed by macrophages or the reticuloendothelial system and thus cannot be degraded and/or removed [76,77]. However, since our CNTs are covalently attached to the underlying materials and the strong mechanical properties of the scaffolds were already validated [61,63], we did not anticipate any associated toxicity. This hypothesis was proven by our studies following the FDA and U.S. Pharmacopoeia recommended guidelines of monitoring toxicity through elution testing, as well as indirect and direct cytotoxicity assessments using “Short” (about 13–16 μm) and “Long” (about 190–230 μm) CNT-coated scaffolds. The current study suggests that varying the CNT length and not necessarily the hydrophilicity or nanoroughness is a key factor in controlling the cellular response of a keratinocyte cell line (HaCaT) on our hierarchical materials. Both hydrophilic and hydrophobic Long CNT-coated scaffolds had significantly lower proliferating cells as compared to hydrophilic and hydrophobic Short CNT-coated surfaces. Long CNTs provide very high surface area, and more random, substantially interconnected structures. Keratinocyte cells may not be able to attach and start ECM secretion, as easily on the longer CNTs due to their randomized orientation and denser clusters of nanotubes. Short CNTs do not completely screen the underlying fiber cloth and do not create randomized clusters of tubes as seen with Long CNTs. This more uniform and less heightened structure of the Short CNTs may be more receptive to cell attachment and ECM secretion as well as promote increased directionality to cells through contact with the weaves of the underlying fiber cloths. Thus, keratinocyte cell growth may be more regulated by available surface area and structural directionality of the CNTs rather than surface wettability. This also suggests that surface nanoroughness may not be a key controlling factor of keratinocyte growth on such surfaces since the covalently-linked Short CNT scaffolds supported keratinocyte cell growth that was similar to the control group. However, the Long CNT-coated scaffolds were more effective for cytoprotection against UVB, which may be due to their increased surface area and screening effect [79]. 

Although these scaffolds require further development and in vivo testing, current studies suggest they could provide a flexible and functionalizable platform for future tissue regeneration applications. This could have applicability as a split-thickness skin graft or as dermo-epidermal substitutes, which can contain multiple cell types and can cover a large area. Bioscaffolds currently in development for wound healing applications have limitations in terms of their sterilization, storage, and proliferative capacities [9]. Keratinocyte cell sheets are too delicate to be transferred by themselves and require a scaffold that can provide mechanical strength as well as biomimetic cues [80,81]. This could be particularly useful for extensive non-healing wounds where robust and non-degradable scaffolds, similar to CNT-coated carbon fiber scaffolds, could provide mechanical strength and cell adhesion properties for tissue retention. Our studies also showed cell migration from the scaffold, which could anchor a graft. However, this needs to be tested in future in vivo studies as the cell lines used in this study were highly proliferative. Additionally, the physiological nature of the tissue may slow down the cell migration from the scaffolds. In our studies, we also attempted to evaluate the exact number of live cells seeded on the scaffold, but the cells grew between and onto the CNT layers making it difficult to attain an accurate measure of cells via trypsinization. Furthermore, the unique advantages described above for these CNTs could be utilized to create a tailored hierarchical scaffold for various tissue regeneration applications. For example, permanent 3D structures could be beneficial in a variety of applications, including tissue regeneration and orthopedic surgical treatments of complex injuries such as osteochondral defects and degenerative tendon ruptures. Studies have shown that CNTs are effective in bone regeneration, repair [82], and increasing bone mineral density in murine models [83]. Overall, the current study supports the biocompatibility of the covalently linked CNT-coated scaffolds and the potential of such multi-scale architecture to support skin cell growth and proliferation for wound healing applications. Additionally, the CNT-coated scaffolds were cytoprotective against UVB exposure and supported cell migration, both from the scaffolds and to the scaffolds, providing further evidence of the possible advantages of using these materials and warrants further investigation for future wound healing applications. 

For any wound healing or tissue engineering application, it is important to assess the potential immunogenicity of the scaffolds. TNF-α is a major pro-inflammatory cytokine and regulates initial inflammation and injury [84,85,86]. Upregulation of TNF-α can also activate the NF-κB pathway resulting in the upregulation of other pro-inflammatory cytokines [87], which could potentially explain the significant increases in IL-6 and GM-CSF in the CFC group. In these materials, the bare CFC control mediated a significant increase in TNF-α secretion from keratinocytes, which appeared to be suppressed by the CNT carpet. Long CNT-coated scaffolds had very little effect on cytokine levels other than a significant increase in IL-4 with hydrophobic scaffolds. Instead, the Short CNT-coated scaffolds induced a cytokine profile more similar to the CFC control but to a lesser degree. The increase in cytokines with the Short CNT-coated scaffolds may be due to decreased screening of the CFC surface by the Short CNTs compared to the Long CNTs and the increased potential for the keratinocytes to interact directly with the CFC. Since CFC used in the study comes with commercial coating, in the future, the composition can be changed, and an uncoated CFC with pure carbon, similar to CNTs, can be used to alleviate the immune response. Lastly, damage to keratinocytes is strongly linked with the release of IL-1α [88], and we did not observe any upregulation of IL-1α in any scaffold groups, supporting the lack of toxicity of these covalently-linked scaffolds. Overall, the Long CNT-coated scaffolds did not modulate the cytokine profile of the keratinocytes, whereas the Short CNT-coated scaffolds increased cytokine secretion. Further optimization of CNT length may yield a scaffold candidate that has optimized cell proliferation as well as migration capabilities along with minimal immunogenic and inflammatory potential.

To expand the immune evaluation, the possibility of B lymphocytes reacting with the scaffolds and inducing antibody production was also assessed. Interestingly, in the CL-01 B-lymphocyte cell line stimulated with T-lymphocyte-derived factors (CD40L and IL-4), all scaffolds increased IgG secretion from these cells, including the CFC controls. In comparison, IgM secretion was only enhanced by the Long CNT-coated scaffolds. However, these effects on antibody secretion were not seen in a different B-lymphocyte cell line (SKW). The SKW and CL-01 B-lymphocyte cell lines are derived from different Burkitt lymphoma patients. However, these cell lines likely originated from different cellular or maturation stages because of their differences in antibody production. CL-01 cells have a basal expression of IgM, which is modestly enhanced by CD40L and IL-4 stimulation, whereas this stimulation does induce a class switch to other antibody classes (i.e., IgG, IgA, IgE) [51]. In contrast, CD40L and IL-4 stimulation of the SKW cells significantly increased IgM secretion but did not induce a class switch to other antibody classes [89]. Modulation of antibody production in the CL-01 B-lymphocyte cell line could have been due to a specific or nonspecific interaction with the scaffolds. B cells are activated when B-cell receptors are cross-linked by foreign particles or antigens. Due to the extensive surface area of CNTs, even a low-affinity interaction between the B-cell receptors and the nanotubes would be sufficient to cross-link the B-cell receptors and activate the B cell to produce antibody, particularly in the presence of T-lymphocyte-derived factors (CD40L and IL-4). Alternatively, the 3D structure of the scaffolds might mimic lymph nodes by bringing the cells closer together and perhaps providing interactions that mimic cell-cell contact and promoting the stimulatory effect of CD40L and IL-4. This might be the more likely case because IgG secretion increased equally in the presence or absence of CNT-coating. These interactions between the CL-01 cells and the scaffolds could also be a result of nonspecific interactions with unique surface proteins, perhaps through potential protein coronas forming on the scaffolds. Additionally, the two B-lymphocyte cell lines (CL-01 and SKW) revealed minimal changes in cytokine secretion due to the scaffolds and therefore is unlikely to account for the altered antibody secretion from the CL-01 cells. Although cytotoxicity and cell growth capabilities of noncovalently CNT-linked supports have been extensively studied, little attention has been paid to the immune response, particularly to B lymphocytes [89,90]. The immune system provides protection against pathogens, but it also plays a significant role in most disease conditions. The current results with a very limited number of immune parameters warrant future studies aimed at comprehensively evaluating the potential of scaffolds to induce or inhibit innate and/or adaptive immune responses.

## 5. Conclusions

In this study, we have successfully created biomimetic carbon fabric scaffolds with the multiscale hierarchy of natural biological materials. Carbon nanotube (CNT) carpets were covalently attached to woven carbon fiber fabric using a two-step technique of surface plasma activation followed by chemical vapor deposition (CVD). These scaffolds were shown to be biocompatible, non-toxic, and suitable for the proliferation of skin cells. Keratinocyte and fibroblast cells migrated effectively from the culture plate toward the CNT-coated scaffolds. The migrated cells continued to proliferate into the CNT carpets, indicating they could support wound healing at the site of injury. Moreover, the scaffolds also provided cytoprotection against environmental stressors such as Ultraviolet B (UVB) rays and seemed to have a minimal impact on immune function and response. Scaffolds with shorter CNT lengths and a hydrophilic surface appear to be the most useful for future commercial development. In summary, this study demonstrates the biocompatibility and potential utility of hierarchical carbon scaffolds that warrant further development for wound healing and tissue regeneration applications.

## Figures and Tables

**Figure 1 nanomaterials-13-01791-f001:**
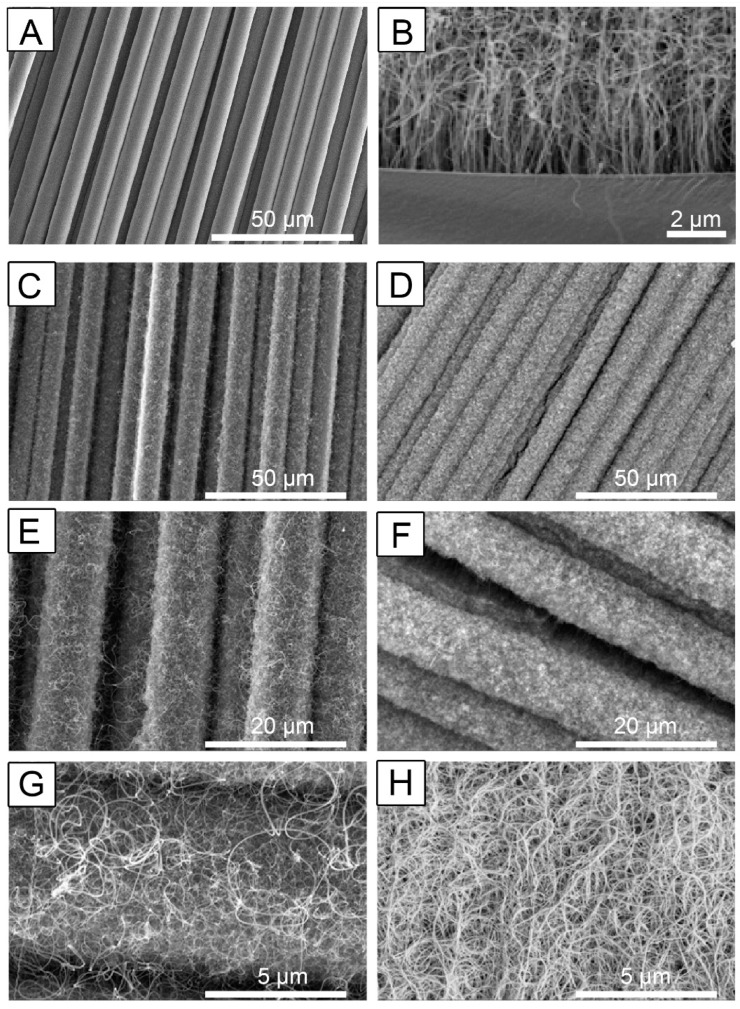
Multiscale hierarchical structure of carbon fiber with Carbon Nanotubes (CNTs). (**A**) is a representative image of fiber cloth without a CNT coating. (**B**) is a representative image showing details of the vertical CNT carpet growth. This image was obtained using a flat graphite surface, where the substrate–CNT interface is visible using a scanning electron microscope (SEM). (**C**,**E**,**G**) are representative scaffold images of carbon fiber cloth with 20 min CNT-coating, and D, F, and H are representative images of the same material with 120 min CNT-coating. Representative scale bars: (**A**,**C**,**D**) = 50 μm; (**B**) = 2 μm, (**E**,**F**) = 20 μm, (**G**,**H**) = 5 μm. The images shown are representative images from three independent samples.

**Figure 2 nanomaterials-13-01791-f002:**
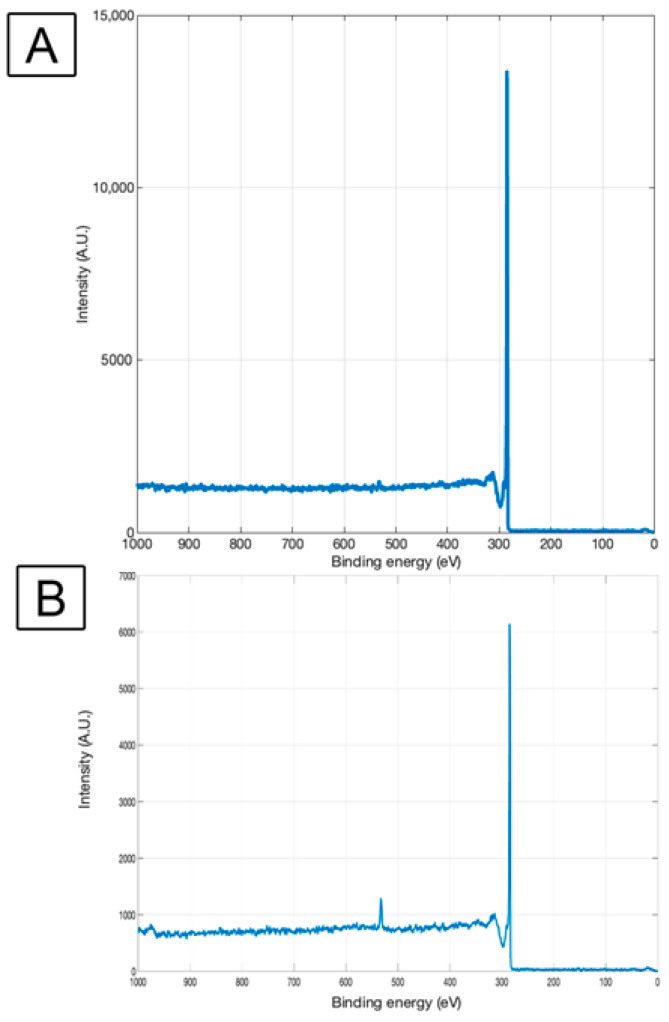
Survey scan of CNT-CFC (**A**) and sodium hypochlorite-based bleach-treated CNT-CFC scaffold (**B**). X-ray Photoelectron Spectroscopy (XPS) survey scans were performed between 0–1000 eV binding energy range to identify elements at the surface. The characteristic peak at 285 eV in both scans represents the presence of carbon in the scaffold. The 533 eV peak in the bleach-treated sample represents the presence of oxygen.

**Figure 3 nanomaterials-13-01791-f003:**
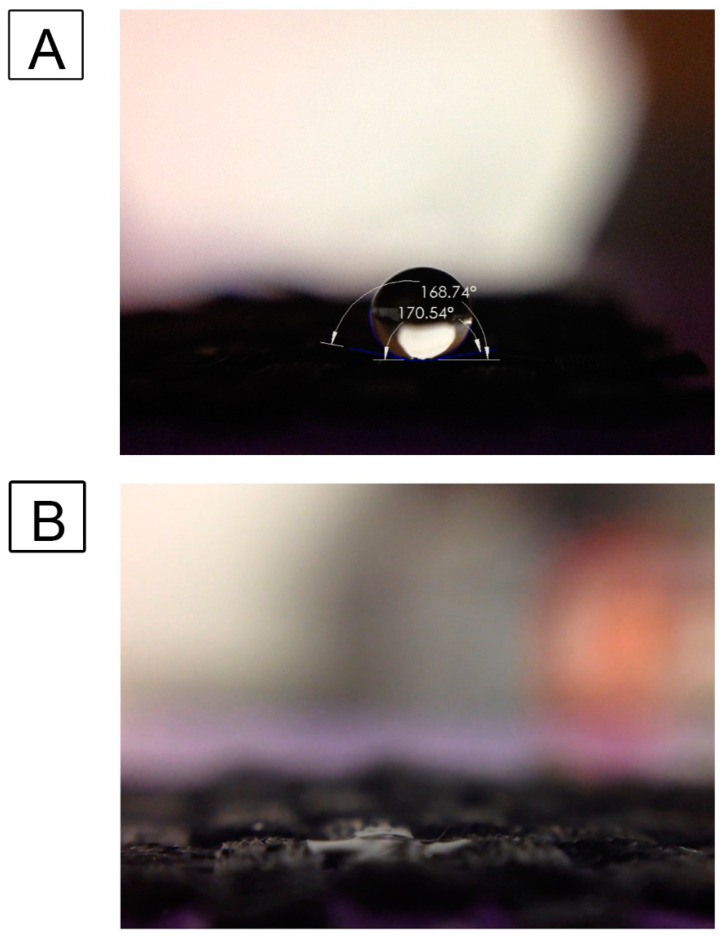
Effect of Sodium hypochlorite-based bleach treatment on water wettability of CNT-coated scaffolds. Water droplet on untreated superhydrophobic CNT-coated fiber surface (**A**) and sodium hypochlorite-based bleach-treated CNT scaffolds (**B**).

**Figure 4 nanomaterials-13-01791-f004:**
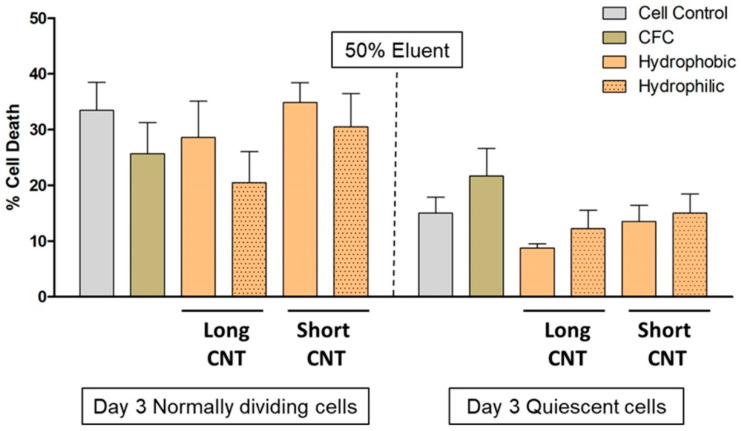
CNT length and wettability do not induce indirect cytotoxicity in HaCaT cells. Extracts were prepared from different scaffolds by incubating scaffolds with Simulated Wound Fluid (SWF) for 14 days, and a 50% extract and complete media mixture were evaluated for potential cytotoxicity on Day 3 on normally dividing and quiescent HaCaT cells by the LDH assay. Scaffolds included CFC, Long CNT-CFC, and Short CNT-CFC with or without sodium hypochlorite-based bleach treatment. The Y-axis represents the quantification of mean formazan-correlated cytotoxicity results normalized to positive cytotoxic controls. The X-axis represents the treatment groups. Statistical significance was measured by two-way ANOVA analysis followed by Bonferroni post-tests. No significant differences compared to the cell control were observed. Results are representative of two independent experiments (n = 3 for each treatment group). Error bars denote SE values.

**Figure 5 nanomaterials-13-01791-f005:**
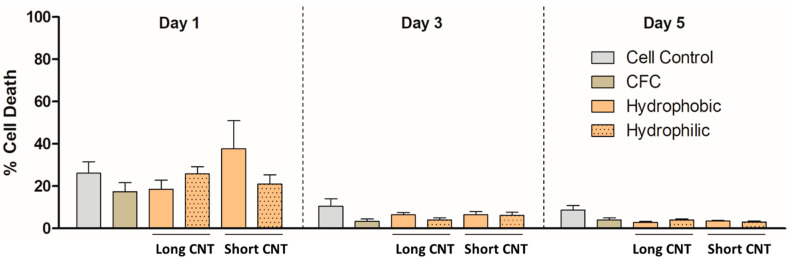
CNT length and wettability do not induce direct cytotoxicity in HaCaT cells. HaCaT cells were grown without any scaffolds (Cell Control) or on different scaffolds for 1, 3, or 5 days and evaluated for cytotoxicity by the LDH assay. Scaffolds included CFC, Long CNT-CFC, and Short CNT-CFC with or without sodium hypochlorite-based bleach treatment. The *Y*-axis represents the quantification of mean formazan-correlated cytotoxicity results normalized to the positive cytotoxic controls. The *X*-axis represents the treatment groups. Statistical significance was measured by two-way ANOVA analysis followed by a Bonferroni post-test by comparing each sample to the corresponding CFC control. Results are representative of three separate experiments (n = 3 for each treatment group). Error bars denote SE values.

**Figure 6 nanomaterials-13-01791-f006:**
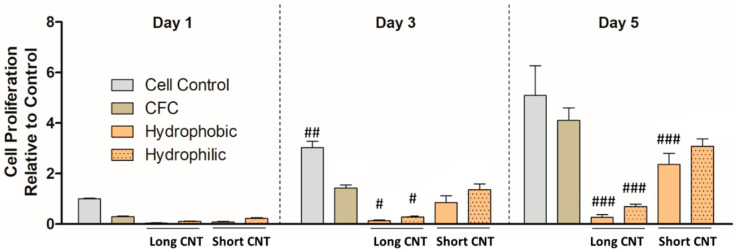
Short CNTs can effectively support HaCaT cell proliferation. HaCaTs were cultured on different scaffolds, and cell proliferation was measured by MTS assay. Scaffolds included CFC, Long CNT-CFC, and Short CNT-CFC with (hydrophilic) or without (hydrophobic) sodium hypochlorite-based bleach treatment. The Y-axis represents the quantification of mean tetrazolium-correlated cell proliferation results normalized to cell control values on Day 1. The X-axis in each graph represents the scaffolds on which the HaCaTs were cultured. HaCaTs cultured on hydrophilic Short CNT-coated scaffolds showed similar cell proliferation as CFC controls. Statistical significance was measured by two-way ANOVA analysis and Bonferroni post-test. Significant differences from the corresponding CFC controls were denoted by #, ##, and ###, which represent significance at *p* < 0.05, *p* < 0.01, and *p* < 0.001, respectively. Results are representative of three independent experiments (n = 3 for each treatment group). Error bars denote SE values.

**Figure 7 nanomaterials-13-01791-f007:**
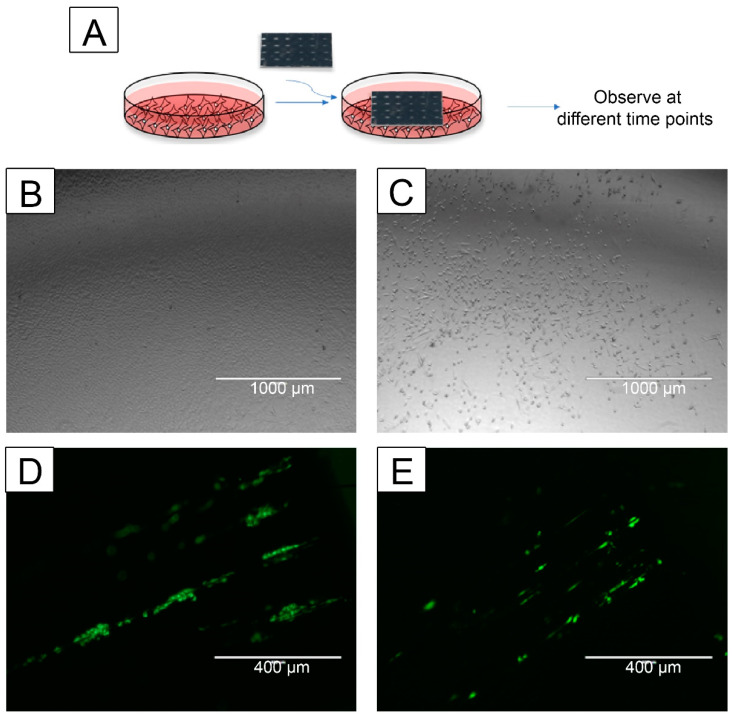
CNT-coated scaffolds support keratinocyte and fibroblast cell migration. HaCaT or NHF1 cells were seeded on the scaffold or tissue culture plate, and cell migration towards an empty plate or scaffold, respectively, was evaluated (**A**). The cell migration from the scaffold towards the plate was observed through phase contrast microscopy on day 15 (keratinocytes) or day 10 (fibroblasts). The cell migration towards the scaffolds was observed through fluorescence microscopy on day 4. Images were acquired using an AMEFC4300 EVOS microscope. Representative images of HaCaT cells (**B**) or NHF1 (**C**) cells migrated from a Short CNT-coated scaffold to the tissue culture plate. Representative fluorescence images of HaCaT cells (**D**) and NHF1 cells that migrated onto the Short CNT-coated scaffolds. Representative scale bars are 1000 μm (**A**,**B**) and 400 μm (**D**,**E**). Images are representative from 3 independent experiments with an n = 3 and 10 images per sample evaluated.

**Figure 8 nanomaterials-13-01791-f008:**
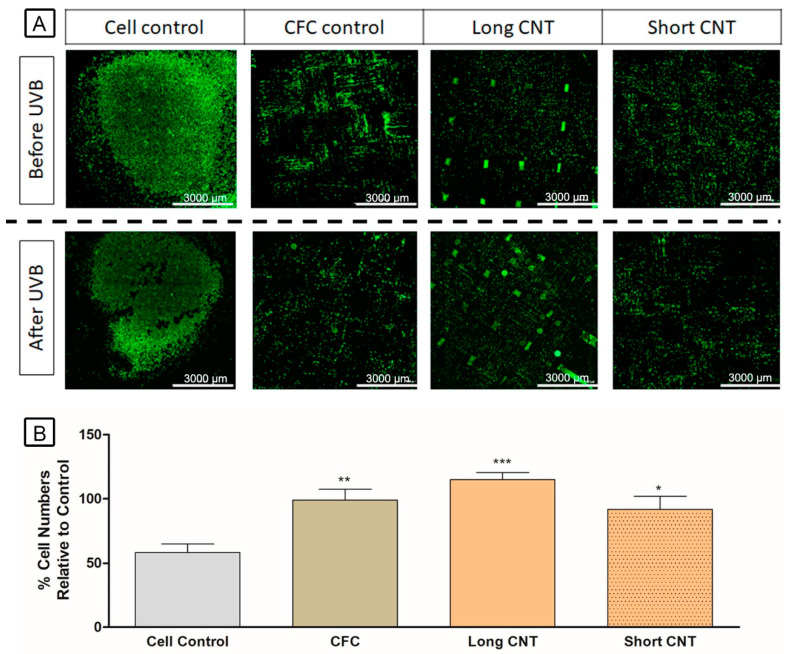
CNT-coated scaffolds provide protection from Ultraviolet-B (UVB) exposure. Cells were labeled with CellTrace™ CFSE, cultured on scaffolds, and incubated for 1 day. Scaffolds were then imaged for cell counts by capturing 6 × 6 stitched images using Agilent BioTek Cytation5. Once imaged, scaffolds were treated with 100 J/m2 UVB and further incubated for 3 days. The scaffolds were again imaged by capturing 6 × 6 stitched images using Agilent BioTek Cytation5. Cell counts were calculated by using the Gen5 software. CNT-coated scaffolds retained a significantly increased number of cells after UVB treatment compared to the cell control, where cells were grown on a cover slide. The upper and lower rows are fluorescent images of cells grown on cover slides and represent cell control, carbon fiber cloth (CFC) control, Long CNT-coated scaffold, and Short CNT-coated scaffold before and after UVB treatment, respectively (**A**). The scale bar represents 3000 μm. Y-axis represents the relative cell number per each group after UVB treatment compared to the cell numbers of the same group before UVB treatment (**B**). Statistical significance was measured by one-way ANOVA with Tukey post hoc analysis. Significant difference from the cell control is denoted by *, **, and ***, which represents significance at *p* < 0.05, *p* < 0.01, and 0.001, respectively. Results are representative of three separate experiments (n = 3 for each treatment group). Error bars denote SE.

**Figure 9 nanomaterials-13-01791-f009:**
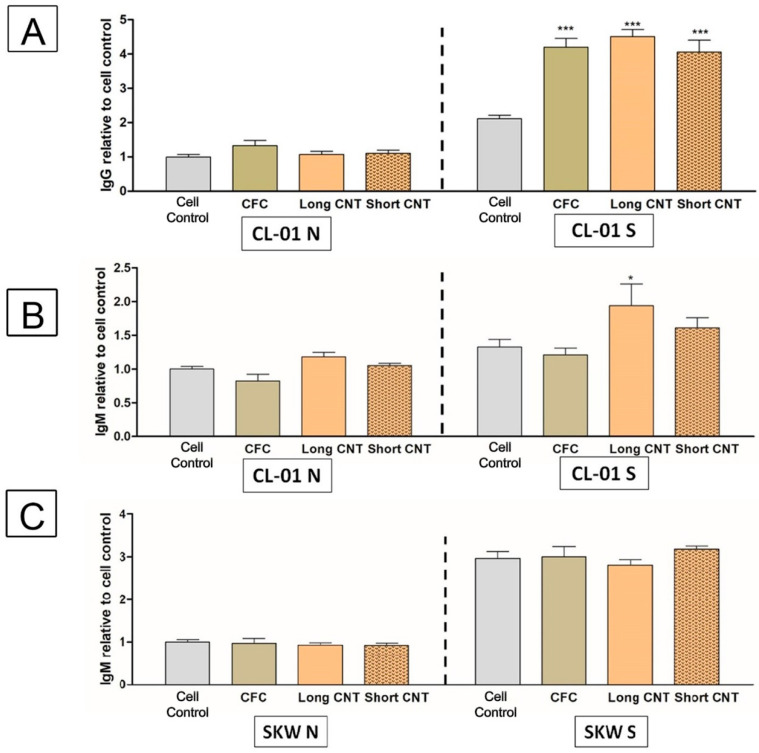
Scaffolds may enhance antibody secretion in stimulated B cells. CL-01 cells (**A**,**B**) and SKW cells (**C**) were incubated for 4 days with or without hydrophilic long or short CNT scaffolds. Levels of IgG (**A**) and IgM (**B**,**C**) were quantified via enzyme-linked immunosorbent assay (ELISA). Supernatants for ELISA analysis were collected from unstimulated (CL-01 N or SKW N) or CD40L + Interleukin-4 (IL-4) stimulated (CL-01 S or SKW S) cells after 4 days of incubation. For each figure, the *Y*-axis represents the antibody levels normalized to the naïve unstimulated cell control (CL-01 N or SKW N). CFC denotes the carbon fiber cloth control. Statistical significance was measured by two-way ANOVA analysis and Bonferroni post-test. Significant differences from the corresponding cell controls were denoted by * and ***, which represent significance at *p* < 0.05 and *p* < 0.001, respectively. Results are representative of three independent experiments (n = 3 for each treatment group). Error bars denote SE.

**Figure 10 nanomaterials-13-01791-f010:**
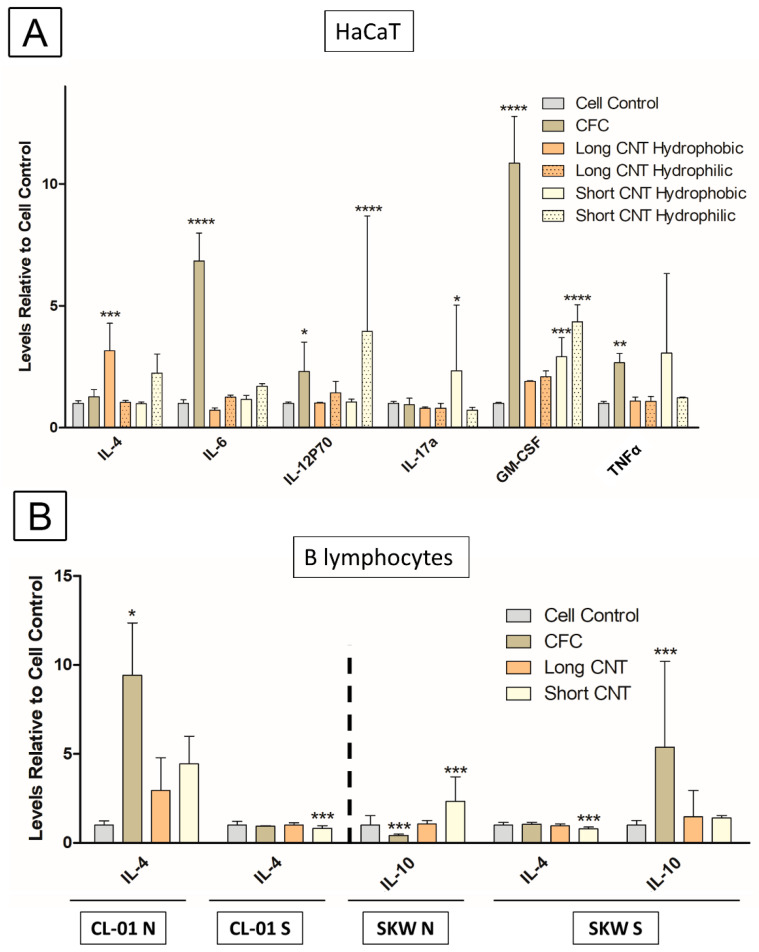
Culturing HaCaT and B cells on scaffolds have modest effects on cytokine secretion. (**A**) HaCaTs were cultured for 5 days on long or short CNT scaffolds, and (**B**) naïve unstimulated CL-01 cells (CL-01 N), stimulated CL-01 cells (CL-01S), naïve unstimulated SKW cells (SKW N), or stimulated SKW cells (SKW S) were cultured for 4 days with long or short CNT scaffolds. CFC denotes the carbon fiber cloth control. Cytokine levels were measured with a Luminex 25-Plex Human ProcartaPlex™ Panel 1B (Invitrogen) assay. Only those cytokines that exhibited any significant changes are shown. The Y-axis represents the cytokine fold change compared to the naïve unstimulated cell control (N). Statistical significance was measured by two-way ANOVA analysis followed by Dunnett’s multiple comparisons test comparing each sample to the corresponding cell control. Significant difference from the cell control is denoted by *, **, ***, and ****, which represents significance at *p* < 0.05, *p* < 0.01, *p* < 0.001, and *p* < 0.0001, respectively. Results are representative of one experiment (n = 3 for each treatment group). Error bars denote SE.

## Data Availability

Not applicable.

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
