# Peer review of "Bioinspired Hierarchical Carbon Structures as Potential Scaffolds for Wound Healing and Tissue Regeneration Applications"

_nanomaterials, 2023, doi:10.3390/nano13111791_

Round 1
Reviewer 1 Report
In this study, bio-inspired hierarchical all-carbon structures comprising CNT carpets covalently bonded to flexible carbon fabric have been investigated for potential wound healing applications.
The major concern is related to the lack of sufficient data to support the conclusion. For example, there were only two images to show the water wettability. However, I think the detection of contact angle can more accurately illustrate this issue.
More detection methods such as WB, immunofluorescence staining for cells should be used to better illustrate protein expression. At the same time, the evaluation of cell survival should also be supplemented with fluorescence maps such as staining of dead cell.
Reviewer 2 Report
Thank you for presenting the above manuscript and selecting me as a reviewer. The manuscript here discusses a novel material that can be used to potentially improve wound healing processes. The study is well structured, but still contains some questions and comments from the reviewer, which should be edited and answered before final publications.
Line 47 continued: surgical intervention should not be presented here as a limited success. It continues to represent the gold standard for many of the skin lesions described above.
Therefore, this point should be clearly weakened here, or it should be clearly presented that surgical intervention is indeed an appropriate therapeutic option that, like all possible options, has advantages and disadvantages. It should be taken into account that the material presented here with the carbon fibers is also far from a significant improvement.
Line 51: Here, too, skin produced in a test tube should not be presented as a savior, since surgical options are still the most commonly used methods. As presented here, no products that have addressed lab-created skin have ever reached a clinical level that would provide a standardized approach to this. This section also needs to be significantly toned down and presented in a way that reflects clinical reality
Line 54: the word biocompatible Is used very vaguely in this context, as is now very common. A clear definition is missing and should certainly be more clearly defined here especially in relation to this study.
Why is the use of CNT scaffolds in the form used here less cytotoxic?
Line 102: what is the evidence for this?
Point 2.10:
There needs to be a correction to the written form here
Discussion: can you authors report in any way on the extent to which the cells used also detach from the carbon fibers used if used in wound healing processes? To what extent could the binding to the used material of the necessary cells be higher than to the necessary biological material? Are there any statements on this regarding detachment or attachment to the material?
What are the differences between nanoscale and microscale carbon fibers in terms of the usage?
Why are the carbon fibers attractive to the cells used?
Line 655: from a reviewer's point of view, I think this statement goes too far. An investigation in terms of live/dead or induction of necrosis or apoptosis with the material used would have been quite useful here. To emphasize biocompatibility and non-cytotoxicity in this way is, in my opinion, going too far!
Round 2
Reviewer 2 Report
Well done revision. Thank you for this.